# The Prevalence of Erectile Dysfunction and Its Association with Cardiovascular Risk Factors in Patients after Myocardial Infarction

**DOI:** 10.3390/medicina57101103

**Published:** 2021-10-14

**Authors:** Egidija Rinkūnienė, Silvija Gimžauskaitė, Jolita Badarienė, Vilma Dženkevičiūtė, Milda Kovaitė, Alma Čypienė

**Affiliations:** 1Faculty of Medicine, Vilnius University, 03101 Vilnius, Lithuania; s.gimzauskaite@gmail.com (S.G.); jolita.badariene@santa.lt (J.B.); vilma.dzenkeviciute@santa.lt (V.D.); milda.kovaite@santa.lt (M.K.); alma.cypiene@santa.lt (A.Č.); 2Vilnius University Hospital Santaros Klinikos, 08661 Vilnius, Lithuania; 3State Research Institute Centre of Innovative Medicine, 08410 Vilnius, Lithuania

**Keywords:** erectile dysfunction, prevalence, myocardial infarction, cardiovascular disease, risk factors

## Abstract

*Background and Objectives*: This study estimates the prevalence and severity of erectile dysfunction and its association with cardiovascular risk factors in patients after a myocardial infarction. *Materials and Methods*: This study included men aged 35–80 years, diagnosed with myocardial infarction and examined in the Department of Preventive Cardiology of Vilnius University Hospital Santaros Klinikos between 2016 and 2020. Anthropometric characteristics, blood pressure, lipid profile, blood glucose levels and prevalence of cardiovascular risk factors were evaluated. The International Index of Erectile Function-5 was used to assess patients’ erectile function. *Results*: A total of 171 patients were analysed. The mean age was 57.6 ± 8.8 years. Of the patients, 42.1, 25.1 and 11.7% had three, four and five established cardiovascular risk factors, respectively. Of the patients, 100% were diagnosed with dyslipidaemia, 90.0%—arterial hypertension, 14.6%—diabetes, 23.3%—smoking, 43.7%—positive familial history and 54.5%—insufficient physical activity. The overall prevalence of erectile dysfunction was 62%. It was scored mild in 37.4%, mild-to-moderate—15.2%, moderate—5.3% and severe in 4.1% of the patients. The mean age was significantly different between severity groups (*p* < 0.001). The study demonstrated a negative correlation between age and total score of the questionnaire (*r* = −0.308, *p* < 0.001). Arterial hypertension was more frequent in the patients with erectile dysfunction (*p* = 0.02). Other cardiovascular risk factors were distributed similarly. *Conclusions*: Erectile dysfunction is common in patients after a myocardial infarction and its severity is age dependent. The prevalence of cardiovascular risk factors is high, with arterial hypertension significantly more frequent in patients with erectile dysfunction.

## 1. Introduction

Erectile dysfunction (ED) is a common male sexual health problem, greatly affecting the quality of life of both men and their partners [1]. In 1993, the National Institute of Health defined ED as a persistent inability to achieve and/or maintain a sufficient erection for successful sexual intercourse [2]. It is estimated that more than 150 million men worldwide have been affected by ED and this number will reach more than 300 million by 2025 [3]. The results of a long-term cohort study of the Massachusetts Male Aging Study showed that some degree of ED occurs in more than half (52%) of healthy 40–70-year-old patients and the prevalence of this health disorder increases with age [4,5,6]. The prevalence of ED varies from 1–29% in men aged 40–49 years to 26–76% in men older than 70 years [6].

Although ED was once thought to be psychogenic in aetiology, nowadays it is mostly associated with vascular disease [7,8]. During sexual arousal, the parasympathetic nervous system is activated and that results in the release of nitric oxide (NO) from adrenergic nerve endings and vascular endothelial cells of erectile tissues [5,6,7,8,9]. Acting through the intracellular cyclic guanosine monophosphate (cGMP) system, NO causes the relaxation of vascular smooth muscle cells, which, in turn, increases blood flow to the corpora cavernosa. Increased internal pressure disrupts venous blood outflow and causes the penis to harden, which is necessary for successful sexual intercourse. In the presence of endothelial dysfunction, NO production is reduced, and it is no longer sufficient to provide the adequate cGMP amount to induce the vasodilatation required to achieve and maintain an erection [9].

There is growing evidence that the formation of atherosclerotic plaques in blood vessels, which is a major pathophysiological mechanism leading to the development of cardiovascular disease (CVD), is caused by functional damage to the inner layer of the vascular wall, the endothelium [10,11]. Endothelial cells serve as a selective barrier of a blood vessel wall and secrete a wide variety of vasoactive substances, such as NO, endothelin-1, thromboxane 2, etc. [12]. Impairment of endothelial integrity increases its permeability, activates platelets and leukocytes, causes activation of various cytokines and reduces NO production, leading to impaired vasodilation, and this point may already be considered the earliest manifestation of the atherosclerosis process [13]. The development of atherosclerosis is understood as a systemic disorder that affects all major blood vessels in the body [10]. Endothelial dysfunction, leading to the onset, further progression of atherosclerosis and subsequent ischemic heart disease (IHD), is also a main mechanism leading to ED in middle-aged men [14]. Thus, this perception allows both, ED and IHD, to be considered as different manifestations of the same disease, atherosclerosis [15].

The Massachusetts Male Aging Study showed that both, ED and CVD, share common risk factors such as arterial hypertension (AH), diabetes mellitus (DM), dyslipidaemia, obesity, lack of physical activity, positive familial history of CVD, smoking and depression [4,16,17,18,19]. In patients with complete impotence, AH has been found two, DM three and CVD four times more often than in the general population, and smoking combined with these risk factors doubles the incidence of ED [4]. The results of a 25-year study showed that age, body mass index (BMI), cholesterol and triglyceride (TG) levels have a significant effect on the risk of developing ED [20]. Meta-analysis showed that nearly half of the men studied with complete impotence were daily smokers, compared with 28% of smoking men in the general population [21]. Obese men with ED have a more significant impairment of endothelial function than obese men without ED [22]. The age range in which most patients with ED fall is a direct risk factor for the development of CVD as well [23].

Several studies indicate that the prevalence of traditional CVD risk factors among patients after a myocardial infarction (MI) is high and at least one modifiable risk factor occurs in more than 80% cases; meanwhile, other sources state that as many as half of the patients do not have any known risk factors for CVD after an MI at all. Overall, patients with a history of MI have a different distribution of traditional risk factors for CVD and their impact on health [24,25,26]. ED has been shown to be significantly more common in patients with multiple risk factors for CVD [27]. The severity of ED also depends on the extent of IHD: patients with MI who have had one coronary artery injury have been found to be less likely to have any difficulties with achieving an erection than patients with two or three coronary artery diseases [28].

The Princeton Consensus Guidelines stated that ED itself is an independent marker of increased risk of CVD, IHD and stroke, which also significantly increases the risk of mortality from all these conditions [18].

In this study, men after an MI were chosen to be studied regarding that ED is a common health disorder both among relatively healthy men with a high number of CVD risk factors and men who have already experienced MI. Montorsi and co-authors found that the incidence of ED among patients after an MI was 42–57% and 10–15% of them were determined with complete sexual dysfunction [29,30]. The higher prevalence of ED after symptomatic heart disease is associated not only with atherosclerosis but also with side effects of beta-blockers, thiazide diuretics and central alpha-agonists often administered after an MI [31,32,33].

In recent decades, it has been observed that traditional risk factors for CVD, particularly AH, dyslipidaemia, obesity, metabolic syndrome (MS) and DM, occur at an increasingly younger age [34,35]. As both CVD and ED share similar pathogenesis and the same risk factors, the relevance of ED and male sexual health also increases.

## 2. Materials and Methods

A retrospective analysis was performed in patients treated in the Department of Preventive Cardiology, VUH Santaros Klinikos, from 2016 to 2020. Detailed inclusion and exclusion criteria for the study is shown in Table 1. A total of 281 patients, who met the inclusion criteria, entered the study. The study protocol was approved by Vilnius Regional Biomedical Research Ethics Committee (Nr. 158200-15-804-316).

All measurements for the study were taken among patients attending regular cardiology visit in the Department of Preventive Cardiology during the period. Physical examination included height; weight; waist circumference; BMI, defined as weight in kilograms divided by height in meters squared; and pulse and blood pressure determination. Blood pressure of the dominant arm was measured in a sitting position after at least 5 min of rest. AH was determined when systolic blood pressure (SBP) was ≥140 mmHg and/or diastolic blood pressure (DBP) was ≥90 mmHg. Normal weight was determined as BMI 18.5–24.9 kg/m^2^, overweight—BMI ≥25 kg/m^2^ and obesity—BMI ≥30 kg/m^2^. Waist circumference of more than 102 cm indicated central obesity.

All participants underwent traditional cardiovascular risk profile analysis (presence of AH, type 2 diabetes, dyslipidaemia, physical activity, smoking and familial history of CVD). AH and diabetes were diagnosed according to the previous medical history of the patient. Data of routinely collected biochemistry blood tests were evaluated. Peripheral venous blood samples were obtained early in the morning after at least a 12-h fast. Total cholesterol, low-density lipoprotein (LDL) cholesterol, high-density lipoprotein (HDL) cholesterol, TG and glucose levels were included into the analysis. Dyslipidaemia was diagnosed when LDL-cholesterol was >1.4 mmol/L, HDL-cholesterol <1 mmol/L, TG > 1.7 mmol/L or patient was using statins. Adequate reduction in LDL-cholesterol levels with statin therapy was determined when LDL-Ch was ≤1.4 mmol/L. Glucose levels greater than 6.1 mmol/L were evaluated as hyperglycaemia. Physical activity was considered insufficient when patients self-reported the frequency and duration of their physical activity as lower than 5 times a week, less than 30 min a day. Smoking status was evaluated as a current smoker, former smoker or non-smoker. Familial CVD history was determined as positive if there was early family history of CVD in first-degree relatives (men ≤55 years and women ≤65 years).

Patients were asked to fill out the 5-item International Index of Erectile Function (IIEF-5) questionnaire for assessment of ED. ED was classified into the following 5 severity levels: score 22–25 indicated none, 17–21—mild, 12–16—mild-moderate, 8–11—moderate and 1–7—severe ED. A total of 171 patients who fully filled out the questionnaire were included into a final analysis of this research.

### Statistical Analysis

IBM SPSS Statistics for Windows, Version 26.0 (IBM Corp., Armonk, N.Y., USA) statistical software was used for data analysis. The data were presented as percentages and means (± standard deviations). The normality of data was determined using Shapiro–Wilk test. Quantitative variables were compared using ANOVA, differences in non-parametric and categorical variables were analysed using Mann–Whitney U and Chi-squared tests accordingly. Spearman’s rank order correlation coefficients were calculated. A *p*-value < 0.05 was considered statistically significant.

## 3. Results

### 3.1. General Characteristics of the Study Sample

The study population consisted of 171 patients who met the study criteria and completed the ED questionnaire. The mean age of the subjects was 57.6 ± 8.8 years. The youngest participant was 35 years old, the oldest—79 years old. A more detailed distribution of subjects by age groups is presented in Table 2.

ED was found in almost two-thirds of the men studied. Mild ED was the most common (37.4%). Slightly more than a third of respondents did not complain of ED (Figure 1).

Dyslipidaemia was detected in 100%, AH in 90% and type 2 DM in 14.6% of the patients studied. Of the 159 patients who reported their smoking status, 23.3% of the subjects were smoking daily and 28.9% quit before or after experiencing MI. The prevalence of CVD risk factors in the study population is presented in Table 3.

SBP ≥140 mmHg was measured in 38% of the patients (*n* = 65). A higher-than-normal BMI was found in the majority (89.5%) of the studied men, of whom overweight was found in 92 (53.8%) and obesity in 61 patients (35.7%). A waist circumference greater than 102 cm was measured in 58.2% (*n* = 92) of all the subjects. Only nine patients (6.0%) had reached optimal LDL-cholesterol levels (LDL-Ch ≤1.4 mmol/L). Hyperglycaemia was detected in 32.2% (*n* = 49) of the subjects. The general characteristics of the study sample are presented in Table 4.

### 3.2. Association between the Prevalence of ED and Its Severity with Age

The mean age was significantly different between the severity groups: the mean age of the patients without ED was 54.2 ± 8.4 years, whereas the patients with any degree of ED were older (mean age 59.7 ± 8.4 years) (*p* < 0.001) (Figure 2). The study demonstrated a negative correlation between age and total score of the ED questionnaire (*r* = −0.308, *p* < 0.001) (Figure 3).

### 3.3. Relationship between ED and Other Risk Factors for CVD

At least one traditional risk factor was identified in all the patients (100%). Three (42.1%) and four (25.1%) established risk factors were the most common. One traditional risk factor for CVD, dyslipidaemia, was diagnosed in four patients. Two risk factors were identified in 31 patients: 87% of the subjects were diagnosed with AH and dyslipidaemia, 13%—dyslipidaemia and smoking. There were no statistically significant differences in the frequency of risk factors between the patients with and without ED (*p* = 0.26) (Figure 4).

The prevalence of all the CVD risk factors examined among the patients with ED and without ED is described in detail in Table 5.

AH was statistically significantly more common in the patients with ED (*p* = 0.02) (Figure 5). There was a significant difference in SBP between the two groups: the mean SBP was 137.3 mmHg in the men with ED, and 130.7 mmHg in the men without ED (*p* < 0.01).

The study groups with and without ED did not differ in the prevalence of dyslipidaemia—no matter what their ED status was, absolutely all the patients after an MI were diagnosed with dyslipidaemia. There was no significant difference in total cholesterol (*p* = 0.73) and LDL-cholesterol (*p* = 0.93) levels between the groups with and without ED.

The prevalence of type 2 DM was similar between the groups (*p* = 0.24) (Figure 6). Glucose concentration also did not differ statistically significantly between the men with and without ED (*p* = 0.16).

## 4. Discussion

Analysis of the data confirmed the assumption that ED is a common health problem in men who have experienced an MI. ED was found among 62% of the patients who visited the Department of Preventive Cardiology after MI. The results obtained are similar to the data of one of the first studies conducted by Montorsi and co-authors, which showed that ED prevalence is 42–57% in patients with CVD [29]. Other studies have reported that the prevalence of ED among such patients varies from 47 to 75% or even in a wider range—24–89% [36,37,38]. This difference in prevalence is explained by the fact that different studies analyse very different populations, define ED differently, have different methods of diagnosing CVD and patients have various co-morbidities and/or take various drugs that may affect erectile function differently [36].

The results of this study showed that the prevalence of ED increases with age. The older the patient is, the more severe the degree of ED is. These results are consistent with many other previous studies examining the association of ED with age [4,39,40,41,42]. The increasing incidence of ED with aging is associated with the process of atherosclerosis, which leads to the development of both symptomatic coronary heart disease and penile vascular arteriopathy [43]. It should be considered that the aging process also causes degeneration of penile tissues, affects central and peripheral nervous system function and older men in general have lower levels of free testosterone in the blood [44,45].

The analysis also showed that patients after an MI had a high prevalence of traditional risk factors: 42% of the subjects had three, 25%—four and 12%—five traditional risk factors for CVD. The most common were dyslipidaemia and AH. All the patients after an MI had at least one risk factor. It is similar to the result obtained in another study that reported that 98% of patients diagnosed with acute coronary syndrome for the first time had at least one risk factor for CVD [25]. The results of our study did not support the conclusion of the United Kingdom Disease Prevention Project that no traditional risk factors were observed in as many as half of the patients with CVD [24].

A high prevalence of risk factors was also observed in the patients after an MI with ED: 39% of men with ED had three, 28%—four and 14%—five risk factors for CVD. The most common were AH and dyslipidaemia (100 and 94% of all the subjects with ED, respectively). The results are significantly higher than those of a similar study by Montorsi and co-authors that analysed 300 patients with acute coronary syndrome: dyslipidaemia was found in 57% and AH in 56% of all the subjects with ED [46]. Based on the results of the study conducted in this paper, the main risk factors for CVD were evenly distributed among patients with and without ED, with the exception of AH, which was statistically significantly more frequent in the group of patients with ED. In the above-mentioned Montorsi study, the distribution of risk factors (AH, type 2 DM, hypercholesterolemia, smoking and obesity) was similar between the ED and non-ED groups, except for type 1 DM, which was statistically significantly more common in the ED group [46].

However, it remains unclear why a significant proportion of men do not develop ED after acute coronary syndrome. In this study, it was found that more than a third of men did not complain of any erectile dysfunction after an MI at all. A similar number of male patients (30%) with diagnosed CVD and no detected ED were obtained in the COBRA study [47]. Although a high prevalence of ED is expected in patients after MI, a significant proportion of the patients do not have ED, presumably due to the different pathophysiology of acute and chronic coronary artery disease. Although atherosclerosis affects both the cardiovascular and penile vessels systemically, patients with acute MI can be expected to have a low prevalence of ED, whereas in chronic coronary heart disease, atherosclerosis has a longer effect on the cardiovascular system and damage of a greater degree can be expected in the pelvic vascular system too [48].

The association between the prevalence of ED and age found in this research remains a major independent risk factor for ED. The results of the above-mentioned COBRA study showed that both in a patients’ group with one, two, or three coronary artery diseases and in a control group, the incidence of ED increased with age, and it was only in the control group that the prevalence of ED increased more slowly, confirming the influence of atherosclerosis ED [47].

Several methodological limitations were encountered in the study. Although the results obtained were consistent with the findings of other similar studies, the study sample was too small to apply the findings to the entire population of patients after MI. Additionally, a control group of patients with ED but without known CVD should be included in the study to better assess the association between ED and traditional risk factors for CVD. A comparison of these groups would lead to more reliable conclusions that traditional risk factors are more prevalent among patients with ED than those without ED. Certain patient data were obtained from previous medical records; therefore, it was not possible to specify the presence of some risk factors. It is also worth mentioning that the International Erectile Function Index questionnaire was used to study the selected population, which is not a very objective research tool, and the number and quality of responses were highly dependent on the respondent. A significant number of eligible patients were not included in the statistical analysis because they refused to complete the questionnaire. This may have been due to the sensitive topic of sexual health, which men tend not to talk about, to hide, or even ignore the problem.

## 5. Conclusions

After analysis of this study sample, it was obtained that ED is a common health problem in men who have had an MI. Men with a history of MI have a high prevalence of traditional risk factors for CVD. AH is more common in men with ED who have experienced an MI than in the group without ED. The study showed that the two most common health problems, ED and IHD, are closely related and these diseases are characterized by a high prevalence of the same risk factors. The frequent occurrence of ED among the patients diagnosed with IHD suggests that erectile function may have been impaired even prior to the onset of MI. This disorder is common in relatively young men; therefore, it is appropriate to interview all patients with traditional risk factors for ED.

## Figures and Tables

**Figure 1 medicina-57-01103-f001:**
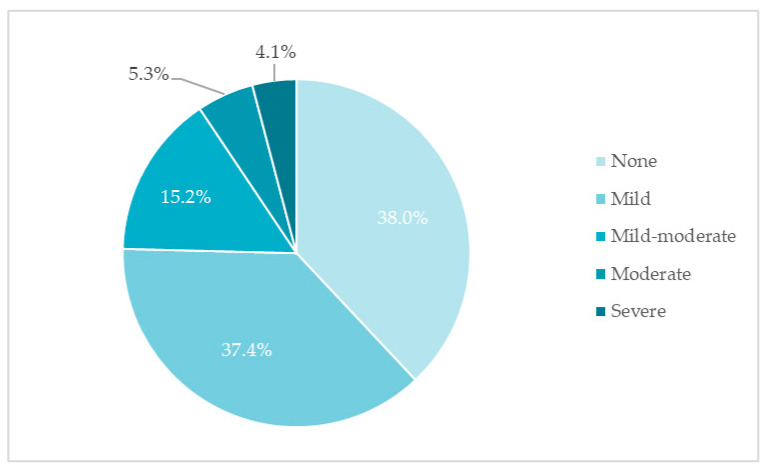
Distribution of erectile dysfunction by severity (%, *n* = 171).

**Figure 2 medicina-57-01103-f002:**
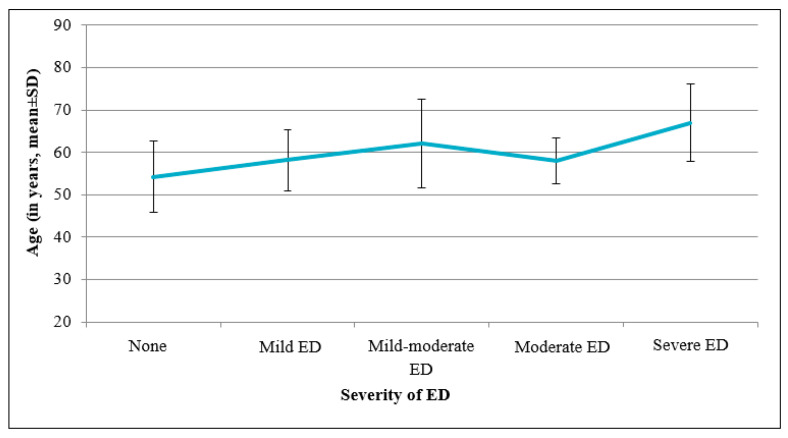
Association between ED severity and patient’s age (mean ± SD, *p* < 0.001).

**Figure 3 medicina-57-01103-f003:**
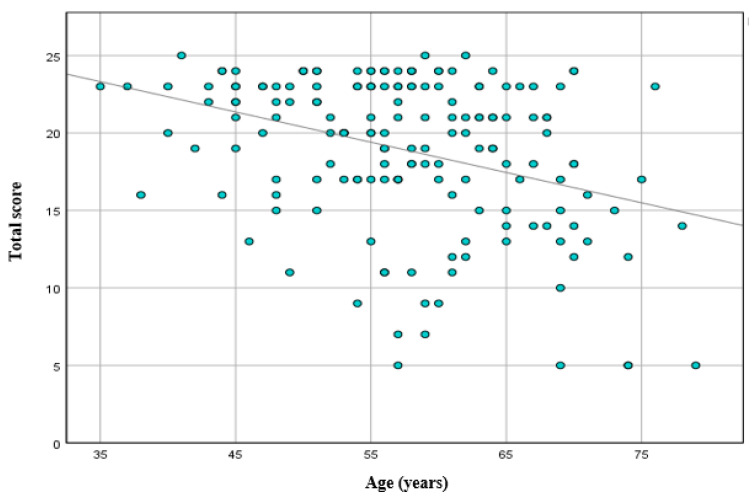
Association between total scores of ED questionnaire and patients’ age (*R*^2^ = 0.124, *r* = −0.308, *p* < 0.001).

**Figure 4 medicina-57-01103-f004:**
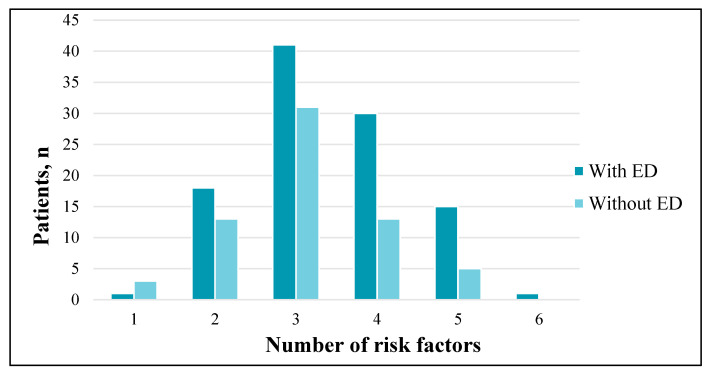
Prevalence of CVD risk factors among patients with and without ED (*p* = 0.26).

**Figure 5 medicina-57-01103-f005:**
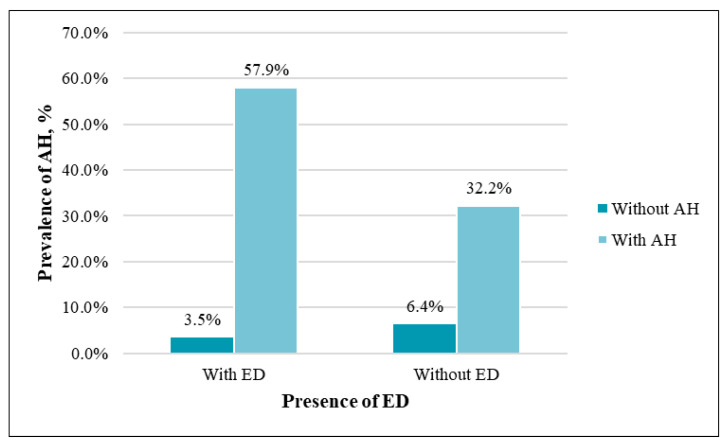
Prevalence of AH among patients with and without ED (*p* = 0.02).

**Figure 6 medicina-57-01103-f006:**
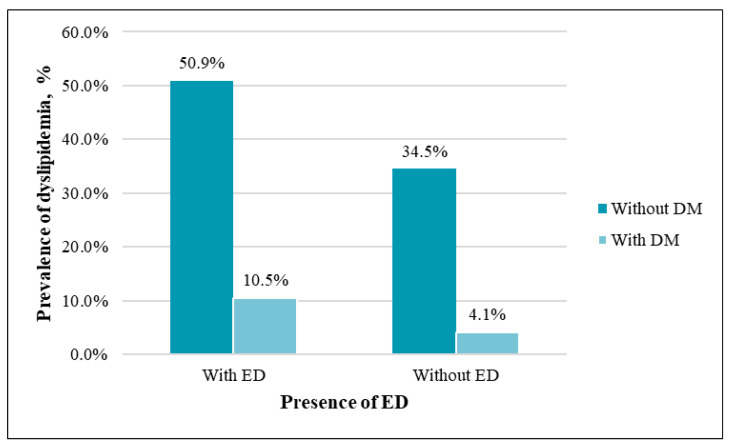
Prevalence of type 2 DM among patients with and without ED (*p* = 0.24).

**Table 1 medicina-57-01103-t001:** Inclusion and exclusion criteria.

Inclusion Criteria	Exclusion Criteria
Male	Female
35–80 years old	<35 and >80 years old
A prior myocardial infarction in the past 3 months to 5 years	A prior myocardial infarction diagnosed less than 3 months or more than 5 years ago
No other known oncological, acute infectious, chronic hematologic diseases, thyroid function disorders, and/or thromboembolic events in the past medical history	Known oncological, acute infectious, chronic hematologic diseases, thyroid function disorders, and/or thromboembolic events in the past medical history

**Table 2 medicina-57-01103-t002:** Distribution of subjects by age groups.

Age Group	Patients (*n*)	Percentage (%)
35–49	32	18.71
50–59	71	41.52
60–69	52	30.41
70–80	16	9.36

**Table 3 medicina-57-01103-t003:** Prevalence of CVD risk factors.

Risk Factor	Total Number of Subjects (*n*)	% of Patients with the Risk Factor Out of Total
Arterial hypertension	171	90.1%
Dyslipidaemia	171	100%
Type 2 diabetes mellitus	171	14.6%
Smoking status: Daily smokers Former smokers Non-smokers	159 159 159	23.3% 28.9% 47.8%
Insufficient physical activity	132	54.5%
Familial cardiovascular disease history	126	43.7%

**Table 4 medicina-57-01103-t004:** General characteristics of the study sample (mean ± SD).

Characteristic	Mean ± SD
Systolic blood pressure (mmHg)	134.76 ± 17.83
Diastolic blood pressure (mmHg)	84.42 ± 10.03
Heart rate (beats/min)	67.17 ± 10.92
Total cholesterol (mmol/L)	4.36 ± 1.13
Low-density lipoprotein cholesterol (mmol/L)	2.57 ± 0.97
High-density lipoprotein cholesterol (mmol/L)	1.12 ± 0.26
Triglycerides (mmol/L)	1.56 ± 1.24
Glucose (mmol/L)	6.03 ± 1.48
Waist circumference (cm)	103.85 ± 10.79
Body mass index (kg/m^2^)	29.13 ± 3.89

**Table 5 medicina-57-01103-t005:** Prevalence of CVD risk factors among patients with and without ED.

Risk Factor (Total, *n*)	Patients with ED, *n* (% of Total with ED)	Patients without ED, *n* (% of Total without ED)	*p*-Value ^1^
Arterial hypertension (*n* = 171)	99 (94.3)	55 (83.3)	*p =* 0.02
Dyslipidaemia (*n* = 171)	106 (100)	65 (100)	*p* = 0.88
Type 2 diabetes mellitus (*n* = 171)	18 (17.1)	7 (10.6)	*p* = 0.24
Obesity (*n* = 171)	41 (39.0)	20 (30.3)	*p* = 0.25
Smoking (*n* = 159)	50 (51.5)	33 (53.2)	*p* = 0.94
Insufficient physical activity (*n* = 132)	45 (56.3)	27 (51.9)	*p* = 0.63
Positive familial history (*n* *=* 126)	39 (50.0)	16 (33.3)	*p* = 0.07

^1^ Reflects the difference in the frequency of risk factors between subjects with and without ED.

## Data Availability

The data presented in this study are available on request from the corresponding author. The data are not publicly available due to privacy issues.

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
