# Peer review of "The Prevalence of Erectile Dysfunction and Its Association with Cardiovascular Risk Factors in Patients after Myocardial Infarction"

_medicina, 2021, doi:10.3390/medicina57101103_

Round 1

Reviewer 1 Report

The objective of the study remains current and despite the small n, which cannot be extrapolated to all patients with erectile dysfunction after myocardial infarction, however, it was evidenced that the classic risk factors for CVD participate in the genesis of ED, therefore the traditional measures and conventional treatment should be established early, which should help a better response regarding sexual function in patients who had AMI.

Author Response

We are grateful to you for considering our work for publication. Thank you for your work and time spent revising our paper.

Reviewer 2 Report

The study is adequately described, easy to follow, and presents a good discussion of findings. Major limitations are appropriately acknowledged in the final paragraph of the discussion section. There are some minor details suggested by this reviewer:

-          It would be interesting to contextualize how the study fits within the introduction. Add a sentence or two at the end of the introduction justifying the necessity of the study.

-          There are a few typos and syntax errors throughout the manuscript.

-          Please include the information about the normality test. Statistics needs a more detailed description.

-          The p-value/ statistical difference is not indicated in the graphs.

Author Response

Dear reviewer,

Thank you for giving me the opportunity to submit a revised draft of my manuscript titled "The Prevalence of Erectile Dysfunction and Its Association with Cardiovascular Risk Factors in Patients after Myocardial Infarction". We appreciate the time and effort that you have dedicated to providing your valuable feedback on
our manuscript.

Here is a point-by-point response to your comments.

  • Comment 1: "It would be interesting to contextualize how the study fits within the introduction. Add a sentence or two at the end of the introduction justifying the necessity of the study"

Response: We have modified the introduction to emphasize this point. You can find changes in the last two paragraphs at the end of the introduction (lines 95-105).

  • Comment 2: "There are a few typos and syntax errors throughout the manuscript."

Response: Thank you for pointing this out. We apologize for our errors and we corrected our typos and other syntax mistakes throughout the manuscript.

  • Comment 3: "Please include the information about the normality test. Statistics needs a more detailed description."

Response: Thank you very much. We added the missing information about normality test. You can see it in line 147. 

  • Comment 4: "The p-value/statistical difference is not indicated in the graphs"

Response: Thank you for pointing this out. We added missing p-values in fig. 4 and 5.

Reviewer 3 Report

This study is a very interesting study. However, followings should be considered:

  1. Please re-check the consistent presence and the proper use of all abbreviations used in the manuscript. (ex. line 79 – “erectile dysfunction” was abbreviated previously in the text, please use the abbreviated form; line 120 - “DTL” it should have been “HDL”)
  2. It is recomanded not to use parentheses within parentheses. If necessary, use brackets inside parentheses. line 114 – “... familial history of cardiovascular disease (CVD)”, CVD was abbreviated previously in the manuscript.
  3. Some minor English problems are present throughout the manuscript. Please re-check the text for grammatical issues.
  4. Please reorganize Table 3. It is difficult to follow: number of patients with risk factor / total number of subjects / (%) of patients with the risk factor out of total. I would recommend presenting the percentage after the absolute number of patients who present the risk factor.
  5. I suggest avoiding the use of abbreviation in the tables, otherwise please define them in the table note, even if they were used previously in the text.
  6. In general, words should be used for numbers from zero through nine, as you did within the manuscript. Try to be consistent in how you treat numbers throughout your manuscript. (ex. line 176 use “one” instead of “1”)
  7. I recommend reporting “p” values larger than 0.01 to two decimal places. Please re-check all “p” values in the text and in tables.
  8. I suggest reorganizing Table 5, since second column appears already in Table 3. To be easier to follow the percentages (Table 5), I recommend introducing the absolute number of patients with ED (n=106) and without ED (n=65).
  9. Conclusions section is missing from the manuscript. I suggest ending the manuscript with a few phrases to conclude the main findings of the study.
  10. Please re-check the references and change where needed according to the citation style guide. (ex. For documents co-authored by a large number of persons (more than 10 authors), you can either cite all authors, or cite the first ten authors, then add a semicolon and add ‘et al.’ at the end. For Journal Articles, journal name and the volume in which the paper was published should be formatted in ITALIC and the year of publication - BOLD.)
  11. It would have been of interest to provide data regarding the long-term use of beta-blockers and the specific molecules in your study population. Since is widely known that an adverse event of most of the molecules is erectile disfunction, your results might have been influenced by the use of certain molecules of beta-blockers.

Author Response

Dear reviewer,

Thank you for giving me the opportunity to submit a revised draft of my manuscript titled "The Prevalence of Erectile Dysfunction and Its Association with Cardiovascular Risk Factors in Patients after Myocardial Infarction". We appreciate the time and effort that you have dedicated to providing your valuable feedback on
our manuscript.

Here is a point-by-point response to your comments.

  • Comment 1: "Please re-check the consistent presence and the proper use of all abbreviations used in the manuscript. (ex. line 79 – “erectile dysfunction” was abbreviated previously in the text, please use the abbreviated form; line 120 - “DTL” it should have been “HDL”)"

Response: Thank you very much for pointing this out. We corrected according to your comment. We corrected the ED abbrevation alone in lines 67, 79, 80, 87, 141, 160, 177, 179, 187, 202, 270. We did the same with other abbrevations (AH, DM, LDL, HDL, TG, etc) throughout the manuscript. 

  • Comment 2: "It is recomanded not to use parentheses within parentheses. If necessary, use brackets inside parentheses. line 114 – “... familial history of cardiovascular disease (CVD)”, CVD was abbreviated previously in the manuscript."

Response: Thank you very much. We used the abbreviation of CVD and eliminated parentheses.

  • Comment 3: "Some minor English problems are present throughout the manuscript. Please re-check the text for grammatical issues."

Response: We apologize for our errors and typos. We have carefully revised our manuscript once again and corrected mistakes throughout the paper.

  • Comment 4: "Please reorganize Table 3. It is difficult to follow: number of patients with risk factor / total number of subjects / (%) of patients with the risk factor out of total. I would recommend presenting the percentage after the absolute number of patients who present the risk factor."

Response: Thank you for your insightful comment. We have modified Table 3. We hope that it is now clearer.

  • Comment 5: "I suggest avoiding the use of abbreviation in the tables, otherwise please define them in the table note, even if they were used previously in the text."

Response: Thank you for your suggestion. We changed abbreviations to full words in the tables.

  • Comment 6: "In general, words should be used for numbers from zero through nine, as you did within the manuscript. Try to be consistent in how you treat numbers throughout your manuscript. (ex. line 176 use “one” instead of “1”)"

Response: Thank you for pointing this out. We corrected this. After changes it is now seen in line 186.

  • Comment 7: "I recommend reporting “p” values larger than 0.01 to two decimal places. Please re-check all “p” values in the text and in tables."

Response: Thank you. P-values are now corrected according your comment.

  • Comment 8: "I suggest reorganizing Table 5, since second column appears already in Table 3. To be easier to follow the percentages (Table 5), I recommend introducing the absolute number of patients with ED (n=106) and without ED (n=65)."

Response: Thank you for your insightful comment. We have modified Table 3 and deleted the same information we used in Table 3. We hope that it is now clearer.

  • Comment 9: "Conclusions section is missing from the manuscript. I suggest ending the manuscript with a few phrases to conclude the main findings of the study."

Response: We added Conclusions section. It can be seen in lines 283-292.

  • Comment 10: "Please re-check the references and change where needed according to the citation style guide. (ex. For documents co-authored by a large number of persons (more than 10 authors), you can either cite all authors, or cite the first ten authors, then add a semicolon and add ‘et al.’ at the end. For Journal Articles, journal name and the volume in which the paper was published should be formatted in ITALIC and the year of publication - BOLD.)"

Response: Thank your for your insights. We changed references according your suggestions.

  • Comment 11: "It would have been of interest to provide data regarding the long-term use of beta-blockers and the specific molecules in your study population. Since is widely known that an adverse event of most of the molecules is erectile disfunction, your results might have been influenced by the use of certain molecules of beta-blockers."

Response: Thank you very much for this comment. Indeed, studies show that the higher prevalence of ED after symptomatic heart disease is associated not only with atherosclerosis but also with side effects of beta-blockers, thiazide diuretics, and central alpha-agonists often administered after MI [1-3].  It would have been definitely interesting to explore this aspect. However, in the case of our study, we could not do that because of missing data about using various medications. However, it is in our plans for the near future to expand our study with data including various medications used after MI.

  1. Adverse reactions to bendrofluazide and propranolol for the treatment of mild hypertension: report of medical research council working party on mild to moderate hypertension. 1981; 318(8246):539–43.
  2. Fogari R, Preti P, Derosa G, Marasi G, Zoppi A, Rinaldi A, Mugellini A. Effect of antihypertensive treatment with valsartan or atenolol on sexual activity and plasma testosterone in hypertensive men. Eur J Clin Pharmacol. 2002; 58(3):177–80.
  3. Sharp RP, Gales BJ. Nebivolol versus other beta blockers in patients with hypertension and erectile dysfunction. Ther Adv Urol. 2017; 9(2):59–63.